

# A novel artificial intelligence-based predictive analytics technique to detect skin cancer

Prasanalakshmi Balaji[1], Bui Thanh Hung[1], Prasun Chakrabarti[2], Tulika Chakrabarti[3], Ahmed A. Elngar[4] and Rajanikanth Aluvalu[5]

[1] Data Science Laboratory, Faculty of Information Technology, Industrial University of Ho Chi Minh City, Vietnam
[2] ITM (SLS) Baroda University, Vadodara, Gujarat, India
[3] Sir Padamapat Singhania University, Udaipur, Rajasthan, India
[4] Faculty of Computers and Artificial Intelligence, Beni-Suef University, Beni-Suef, Egypt
[5] Department of IT, Chaitanya Bharathi Institute of Technology, Hyderabad, India

## ABSTRACT

One of the leading causes of death among people around the world is skin cancer. It is critical to identify and classify skin cancer early to assist patients in taking the right course of action. Additionally, melanoma, one of the main skin cancer illnesses, is curable when detected and treated at an early stage. More than 75% of fatalities worldwide are related to skin cancer. A novel Artificial Golden Eagle-based Random Forest (AGEbRF) is created in this study to predict skin cancer cells at an early stage. Dermoscopic images are used in this instance as the dataset for the system's training. Additionally, the dermoscopic image information is processed using the established AGEbRF function to identify and segment the skin cancer-affected area. Additionally, this approach is simulated using a Python program, and the current research's parameters are assessed against those of earlier studies. The results demonstrate that, compared to other models, the new research model produces better accuracy for predicting skin cancer by segmentation.

## INTRODUCTION

Skin cancer is brought on by malignant tumors, which develop when the proliferation of abnormal skin cells is left unchecked (*Nandhini et al., 2021*). The majority of cancer cases are caused by unprotected skin being exposed to UV radiation (*Reichrath, 2020*). Nowadays, 95 percent of skin malignancies diagnosed are squamous or carcinoma and 5 percent are melanomas (*Daniels et al., 2020*). Due to numerous factors like infections, allergies, and exposure to the sun, skin alterations happen making it the main cause of skin cancer. Skin cancer is one of the cancers that has been diagnosed and has led to the death of millions of people in recent years (*Kassem, Hosny & Fouad, 2020*). Melanoma and eczematoid are two examples of skin malignancies. Typically, skin tumors that develop on the epidermis are a common condition (*Boutry et al., 2021*). Melanoma and eczematoid are two examples of skin malignancies. Typically, skin tumors that develop on the epidermis are a common condition (*Boutry et al., 2021*). Apart from the fact that most skin tumors in the globe are benign and hardly ever develop into skin cancer, there are also dangerous

Corresponding author
Rajanikanth Aluvalu,
rajanikanth.aluvalu@gmail.com

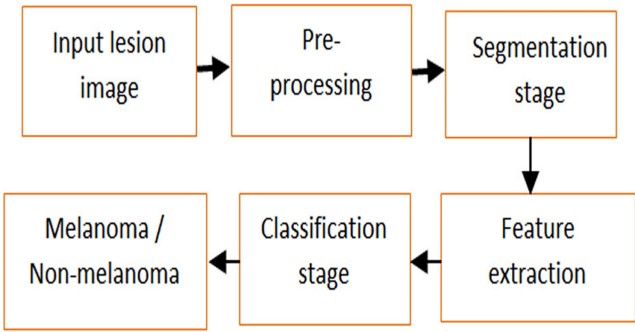

**Figure 1 Analysis and detection of skin lesion.**     

malignant skin tumors that, if left untreated, can kill the host by spreading throughout the body (*Vardasca & Magalhaes, 2022*; *Kassem, Hosny & Fouad, 2020*). To lower the fatality rate, early detection and timely treatment are hence more crucial (*Poggiali et al., 2020*). More than 125,000 new cases of melanoma are diagnosed each year, while there are 3 million non-melanoma cases (*Zhang et al., 2019*).

More often, both melanoma and non-melanoma skin tumors may lead to death of people (*Mühr, Hultin & Dillner, 2021*). As a result, squamous cell carcinoma and basal cell carcinoma attack non-melanoma skin (*Liu-Smith, Jia & Zheng, 2017*; *Alagu & Bagan, 2021*). Melanoma causes aberrant melanocyte proliferation, and the cells that produce pigment also give skin its color. A larger probability of curing melanoma exists with early identification. However, it expanded to other body areas and had permanent effects (*Antunes et al., 2020*). Overfitting, low detection accuracy and error rate are further issues with a skin cancer diagnosis (*Nahata & Singh, 2020*). Large-scale data training for disease prediction is the main advantage of utilizing a deep learning model in medical picture analysis (*Chen et al., 2019*). Figure 1 defines the steps of investigation and detection of skin lesions.

Machine learning methods are chosen in the proposed approach since, compared to the Deep Learning (DL) model, the Machine Learning (ML) technique achieves greater disease prediction accuracy due to the hyper parameter tuning (*Galdran, Carneiro & González Ballester, 2021*). Additionally, by preventing data overfitting during training, labeled data can lower training errors (*Bilbao & Bilbao, 2017*; *Alom et al., 2019*). Additionally, various rule-based techniques make it simple to solve the machine learning model (*Zhou et al., 2020*). Since several decision tree layers are used to process the ML model's function. The memory and control mechanism in the contemporary ML allows it to carry out the procedure better (*Botvinick et al., 2019*). To segment the damaged section and control the degree of disease by early-stage prediction, filtering technique (*Rahman et al., 2022*), automatic and optimal pipeline approach (*Wei et al., 2021*), *etc*. The suggested approach is capable of identifying and classifying the area affected by skin cancer. The main aim of the designed model is to enhance the prediction results of skin cancer by accurate segmentation and classification.

Below are a few current literature reviews that focus on skin cancer prediction. A rectification and uncovering of ML for the diagnosis of skin cancer were developed by

*Nauta et al. (2022)*. Dermoscopic images are initially trained using a typical VGG16 classifier on the public ISIC dataset, which consists of colored patches and elliptical charts. As a result, the created technique automatically eliminated the patches that were present in the trained images and measured changes as they occurred during prediction; however, prediction accuracy was low when compared to other strategies.

It is more difficult to detect similarities and weak contrast between lesions and skin. *Rahman et al. (2022)* suggested a method for filtering the dataset's noise using dermoscopic images like anisotropic diffusion. Additionally, the affected region of the skin cancer is segmented using the quick bounding box technique. Additionally, the created technique performs better; however, segmenting the skin cancer-affected region took longer.

An innovative DL method for identifying the type of cancer depending on its phases was put forth by *Mijwil (2021)*. Three architectures were employed to determine the precise classification of images. Additionally, the evaluated high-resolution images achieved sensitivity and specificity rates of 86.14 percent and 87.66 percent, respectively, although the error rate is significant due to vast data. Early identification and detection of skin cancer result in a high level of effort in disease management.

Using dermoscopic images, *Wei et al. (2021)* suggested an automatic and ideal pipeline technique for the diagnosis of skin cancer. It uses the threshold technique to characterize the Region of Interest (RoI) and noise reduction. The issue of offloading arises when comparing existing ways to test reliability.

A unique ML framework for the diagnosis of skin cancer was created by *Murugan et al. (2021)*. In addition, skin cancer is the most prevalent disease in people and is determined by skin tone. First, apply a median filter to the skin image, then use a mean shift to segment the affected area. After that, features are retrieved and then classified using a support vector machine (SVM). Additionally, the segmentation of the lesion area can be done with high precision, although the false detection rate is significant.

The key roles of this proposed model are summarized as;

- Initially, the dermoscopic image dataset was collected and trained in the system using the python tool.
- Subsequently, a novel Artificial Golden Eagle-based Random Forest (AGEbRF) model was developed to predict skin cancer by segmenting the affected region.
- Hereafter, the fitness of the golden eagle is updated in the Random Forest (RF) that is used to segment the affected part images.
- Hence, the developed AGEbRF segments the affected part of skin cancer with high performance.
- Finally, the segmented images are utilized to track the dataset for predicting the possibility of skin cancer.
- Moreover, the developed approach is validated using recent prevailing approaches in terms of detection accuracy, specificity, sensitivity, F-measure, and precision.

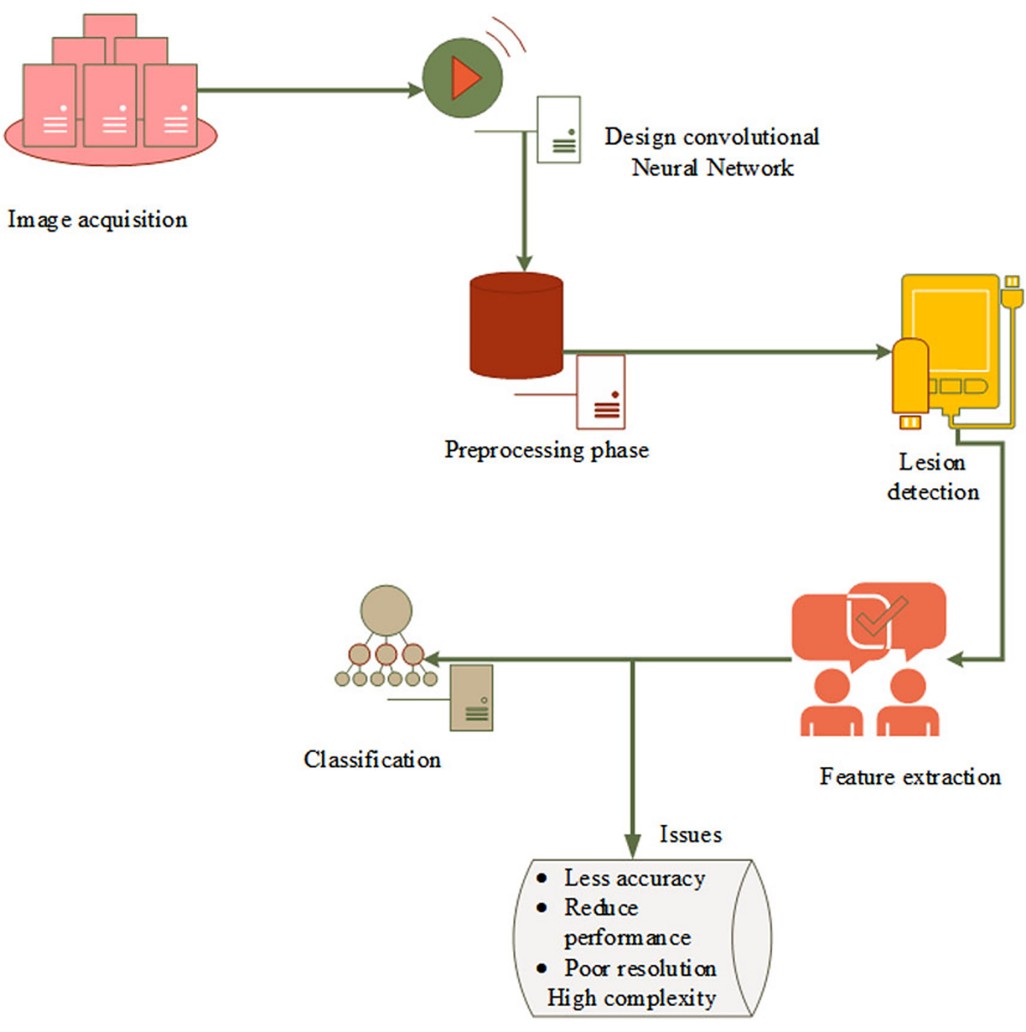

**Figure 2  Basic model and problem definition.**     

## MATERIALS AND METHODS

There are 200 different forms of cancer around the globe, and they all disrupt tissue, disrupt organ development, and penetrate cells. Tumors are frequently produced cells and spread to other tissues. Additionally, melanoma has pigment cells that result in a broad, dark lesion. There are five phases in skin cancer detection, including preprocessing, lesion identification, feature extraction, and classification.

Figure 2 depicts the fundamental model of the system and research problem. First, capture the dermoscopic images and then do preprocessing to reduce noise and enhance image quality. The original image is then masked using the threshold value, and the border, color, asymmetry, and diameter of each feature are processed. The classification layer, which is the last to detect skin cancer, has a few drawbacks, including high complexity, misclassification, poor resolution, and lower detection accuracy. However, as skin cancer segmentation is handled manually, there is a high possibility of incorrect classification and

errors as well as erroneous predictions. This creates a revolutionary machine learning approach to identify and divide the skin cancer-affected area.

Identification of the damaged area is more important for the treatment of skin cancer because it is impossible to finalize treatment measures without the knowledge of the affected area. The current study has proposed an Artificial Golden Eagle-based Random Forest (AGEbRF) for the exact prediction of the damaged skin areas to overcome this issue. This method updates the golden eagle fitness in the classification layer to improve segmentation effectiveness. Consequently, it uses dermoscopic images to segment and predicts the area affected by skin cancer. Finally, the results of the intended strategy are contrasted with those of other strategies. Figure 3 displays the proposed methodology.

## Data acquisition

Multiple dermoscopic images were gathered since dermoscopic skin scans are necessary for identifying illnesses. For training and testing purposes, a dermoscopic image dataset obtained from the internet was used in this study. The skin cancer dataset (*Codella et al., 2018*; *Tschandl, Rosendahl & Kittler, 2018*) was utilized, which also has more dermoscopic images of patients. The dataset used in the study includes 10,015 dermatoscopic images for training from Task 3 of the dataset and 1,512 images from the same task for testing purpose contributing to a total of 11,527 images. As a result, both benign and malignant skin lesions are mixed in the dataset. It consisted of around 53.98% of male patients, 45.45% of female patients and an unknown category of 0.57%. The images are also used to teach the system; Table 1 elaborates on the testing and training dataset. The dataset was found to be biased in terms of the types of lesion, even then since the article concentrates on just the prediction process, we do not concentrate on the bias since the bias in category of lesion does not affect the lesion prediction results.

## Process of AGEbRF to segment the affected part

In this approach, the AGEbRF mechanism is developed for segmenting and predicting skin cancer. The developed GEbRF is the combined form of the Golden Eagle Optimization (GEO) algorithm (*Mohammadi-Balani et al., 2021*) and random forest (RF) (*Lee et al., 2022*).

The behavior of the GEO function is initiated in the RF detection layer which can enhance disease prediction. In this update, the fitness of the golden eagle enhances the tracking results of the affected parts of skin cancer. It searches the stack by spiral trajectory and indicates the propensity to cruise in the primary stage, which identifies the attack and provides prey information to the other eagles it enhances the prediction results. In the current study, the GEO function is used to pinpoint the exact location of the disease in the skin. Initially, the system is trained on the dermoscopic image dataset. The random forest has been accomplished by numerous sub-trees that contain preprocessing, feature extraction, segmentation, and prediction.

### Pre-processing and feature extraction

The dermoscopic image dataset is fed into the input layer, and the noise and errors in the dataset are removed in the following layer. Furthermore, the input layer performs pre-

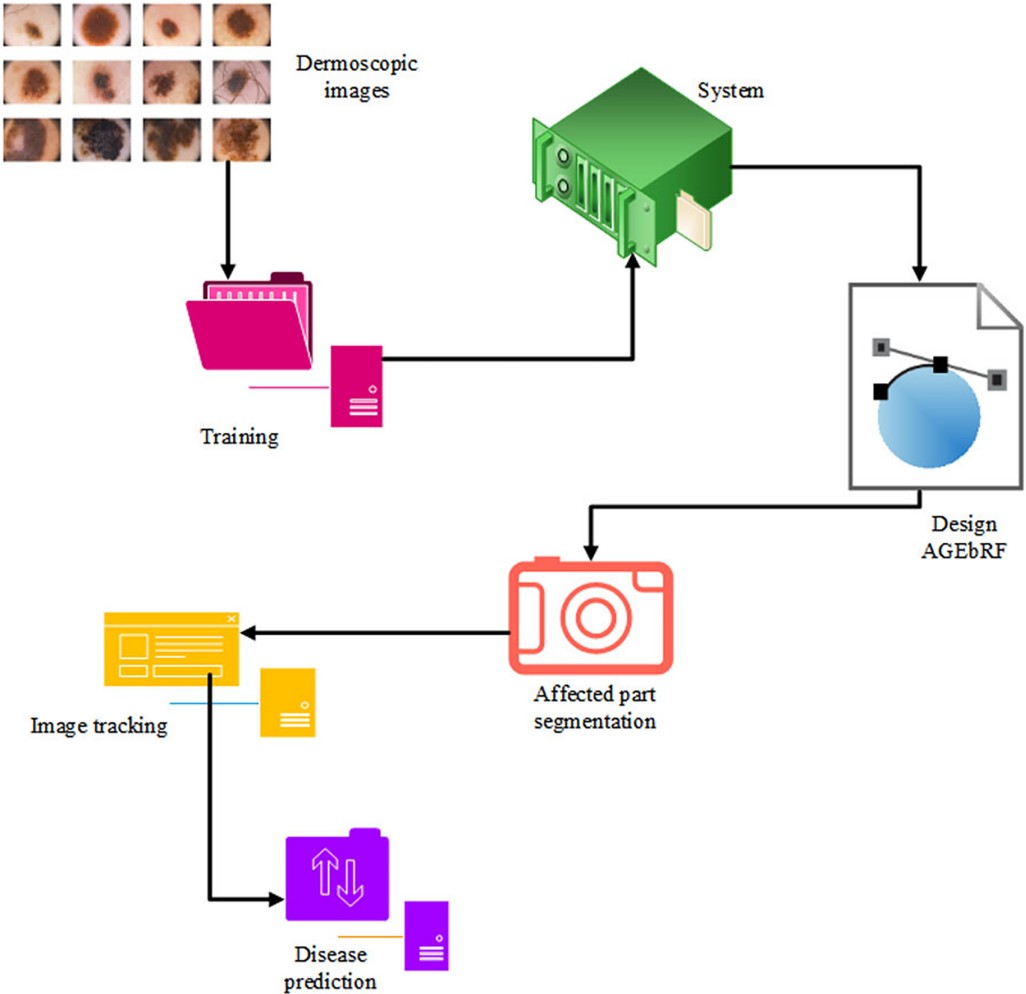

**Figure 3 Proposed workflow.**

**Table 1 Dataset description.**

| Training | Nos | Testing | Nos |
|---|---|---|---|
| Benign | 1,290 | Benign | 500 |
| Malignant | 1,307 | Malignant | 500 |
| Total training dataset | 2,597 | Total testing dataset | 1,000 |

processing and feature extraction processes on dermoscopic skin images. Then, using a non-linear operation, features are extracted from dermoscopic images based on shape, color, size, and texture. In Eq. (1) the extraction process non-linear operations are mentioned,

$$S_i d = f(L(n) * S_i s) \tag{1}$$

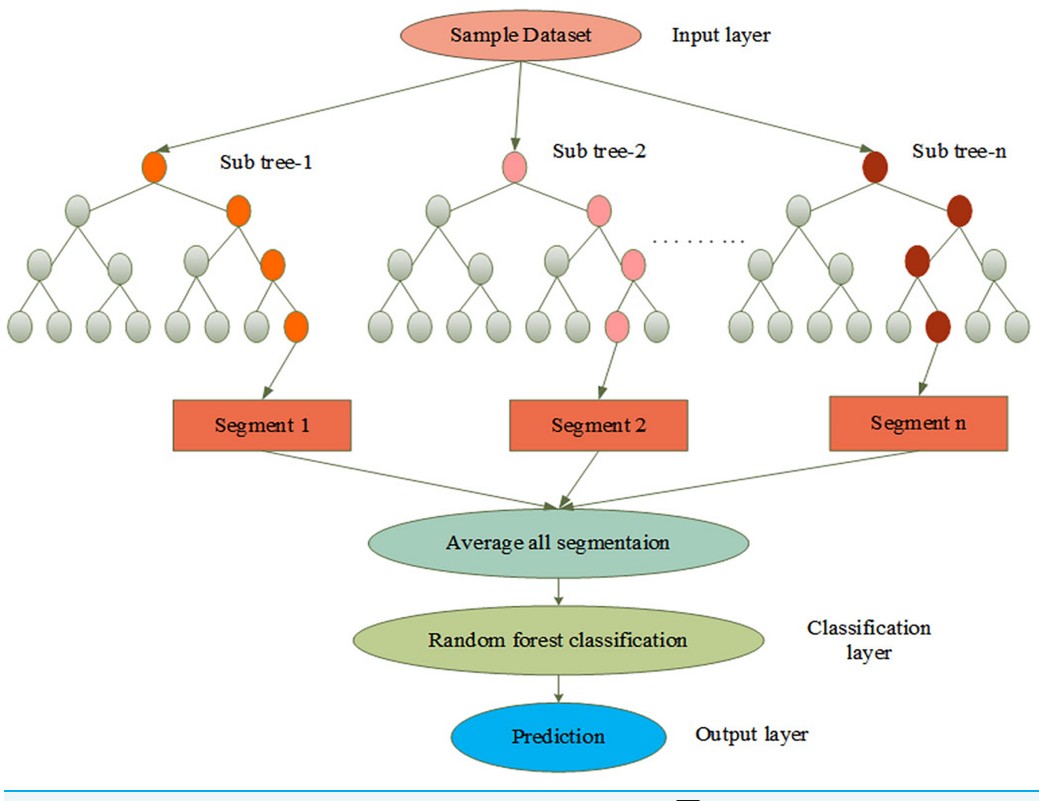

**Figure 4  Process of AGEbRF network.**   

Let, $S_i s$ is denoted as the input image, $S_i d$ is represented as the extracted output, and non-linear operation is represented as $L(n)$. The process of AGEbRF is represented in Figure 4.

### *Segmentation and prediction*

The extracted output is passed to the classification layer, which performs the classification process. The golden eagle fitness function is used to improve classification in this layer. It reduced the spatial resolution to achieve spatial invariance to falsification and input translations. Furthermore, this layer classifies patients with skin cancer and determines whether it is benign or malignant. After that, the classified output is sent to the prediction layer, which can segment the affected part. As a result, the segmentation and classification processes are carried out using AGEbRF, which is represented in Eq. (2),

$$S_i d^{c_l} = f\left\lfloor \sum\nolimits_{k \in A_l} S_i d^{c_l - 1} * KN_{k_l}^{c_l} Srb_i + A_l^{c_l} \right\rfloor dr_{iter+1} \tag{2}$$

Let, $c_l$, represent classification layer, $c_l - 1$ is denoted as the sub-tree segmentation, $S_i d^{c_l - 1}$ is considered as the input characteristics of this layer. Moreover, kernel maps and additive bias are denoted as $KN_{k_l}^{c_l}$ and $A_l^{c_l}$. Furthermore, $dr_{iter}$ is considered as the iteration value, $Srb_i$ which is represented by the fitness of the golden eagle. Additionally, $A_l$ is denoted as selection input maps. $k$, and $l$ are represented as input and output. Finally, the output layer presents the segmented output and uses a decision tree built from numerous

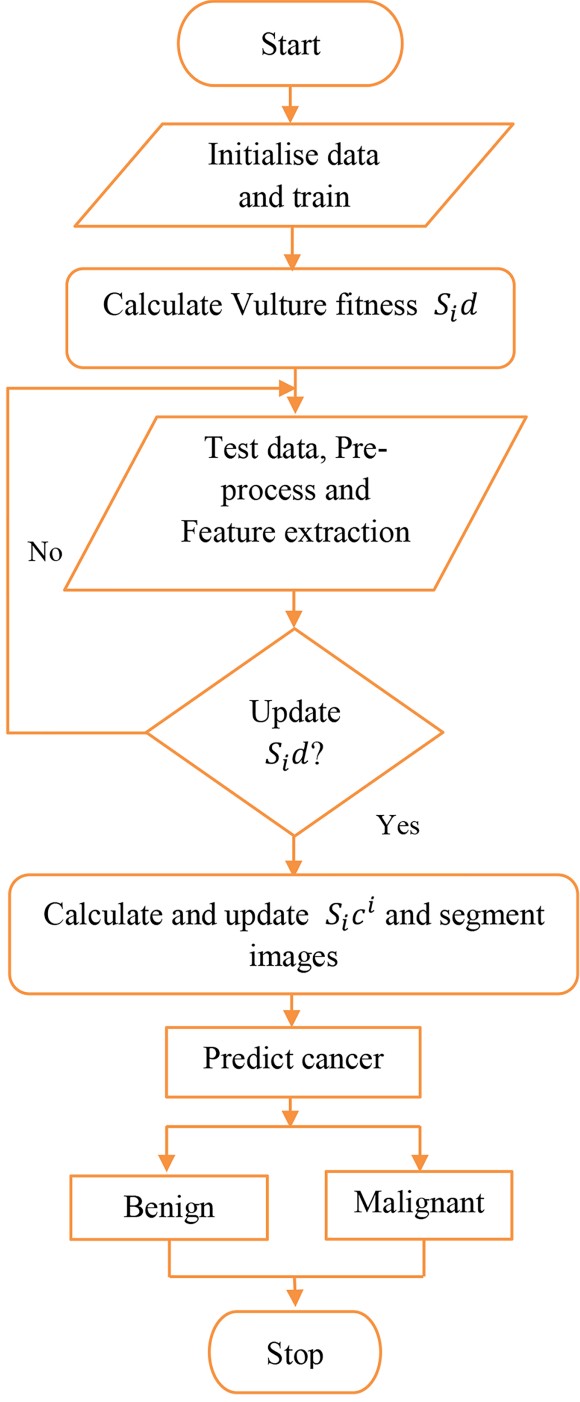

**Figure 5 Flowchart of proposed AGEbRF approach.**

samples to predict the skin illness as benign or malignant. Here, the affected part is divided into categories using the AGEbRF method. Hence, the dermoscopic image dataset was segmented, and predict the damaged portion of skin cancer using the created AGEbRF method. A flowchart of the created AGEbRF is also provided in Figure 5.
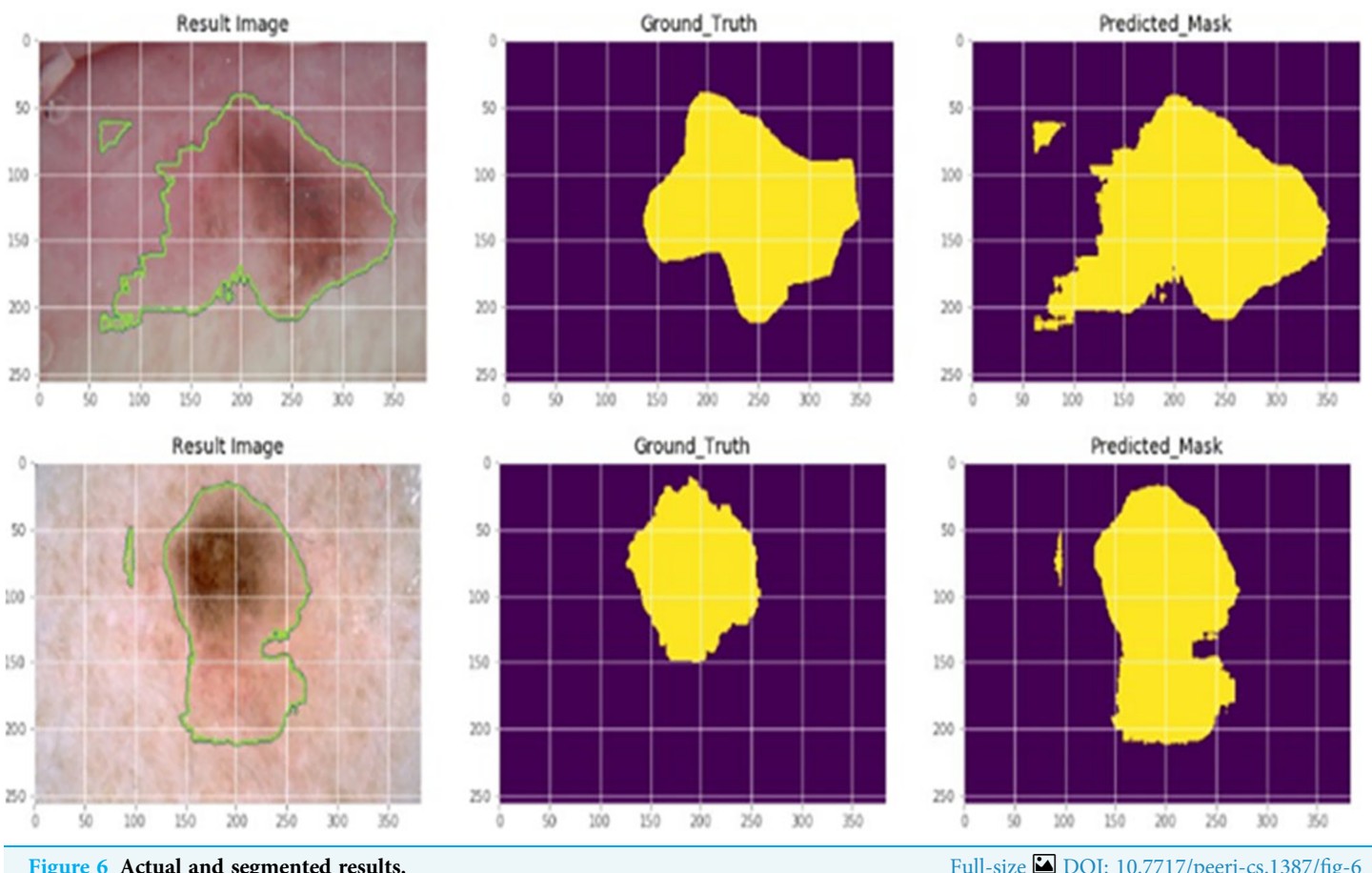

**Figure 6 Actual and segmented results.**     

Additionally, the dataset for determining the probability of skin cancer, whether it is benign or malignant, is predicted using the segmented images.

## Case study

Skin cancer is typically regarded as the most dangerous type of cancer. Periodically, early detection is required, and a cancer diagnosis must be precise and effective if people are to be saved. The suggested AGEbRF model is first applied to the dataset once the dermoscopic dataset has been trained to the system. Additionally, the pre-processing and feature extraction functions are used in the dataset to eliminate unintentional image flaws. The feature extraction technique, which uses the suggested model, extracts the pertinent features from the filtered input images. These attributes include shape, size, color, and texture. Figure 6 provides specifics on the sample dermoscopic images and the expected outcomes.

The classification layer of the suggested model is then used to segment the impacted portion. The suggested AGEbRF technique has segmented the affected area of skin cancer. The generated model was then used to assess the possibility of skin cancer.

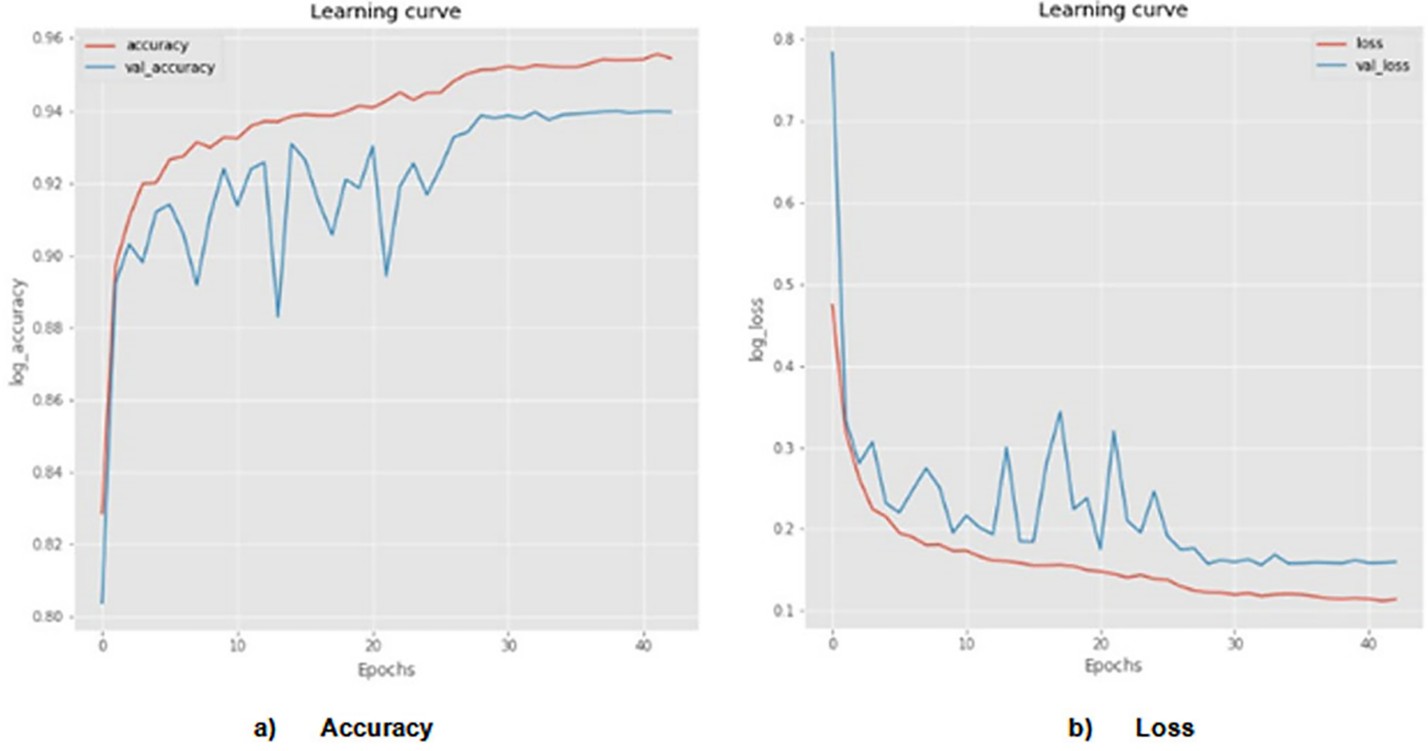

**Figure 7 Actual and segmented results.** Accuracy and loss of actual and segmented image.

This uses the segmented output as the test to predict potential skin cancer cases in a given dataset. Finally, the probability of skin cancer is determined using the segmented images. Here, the dataset images are regarded as the testing images while the segmented images are used as input images. In these images, the created AGEbRF model is used to assess the possibility of the disease, whether benign or malignant.

The probability of skin cancer is 50 percent benign and 50 percent malignant when utilizing the proposed AGEbRF model. Here, the input images are taken into consideration as 1,000 numbers, of which 500 are malignant and 500 are benign. As a result, the suggested model pinpoints the skin cancer's affected area and Figure 7 detailed the comparison of accuracy and loss of established framework.

The designed AGEbRF technique is implemented using a Python program, and several metrics like accuracy, sensitivity, specificity, F1-measure, and precision are measured. Additionally, the developed approach is verified using established techniques such as hybrid feature fusion and ML (HFF) (*Rahman et al., 2022*), skin cancer classification (SCC) (*Mijwil, 2021*), modified thermal exchange (MTE) (*Wei et al., 2021*), and diagnosis of skin cancer by ML (DSC) (*Murugan et al., 2021*), region extraction and classification of skin cancer (RCSC) (*Saba et al., 2019*) and the ensemble lightweight technique (ELT) (*Wei, Ding & Hu, 2020*).

### Accuracy measurement

The degree of measurement of the proposed model's working efficiency is known as accuracy. Additionally, Eq. (3) expresses the proportion of correctly expected observance to all observations.

$$A_c = \frac{t_p + t_n}{t_p + t_n + f_p + f_n} \qquad (3)$$

where $t_p$ is denoted as the true positive of accurate prediction and precise segmentation, $t_n$ is represented as the true negative of accurate prediction and incorrect segmentation, $f_p$ is denoted as the false positive of inaccurate prediction and exact segmentation, and $f_n$ is considered as the false negative of inaccurate prediction and incorrect segmentation of affected skin part.

Using established techniques including HFF, SCC, MTE, DSC, and RCSC approaches that resulted in values as listed in Table 2, the correctness of the proposed AGEbRF model is estimated and validated.

The new strategy has a higher accuracy value of 99.96% than other ways, but the existing approaches have achieved lesser accuracy, demonstrating the usefulness of the proposed approach. Figure 8 displays a comparison of accuracy measurement results.

### Sensitivity measurement

The amount of accurately anticipated true positives is determined using sensitivity. Additionally, Eq. (4) measures the probability of predicting the skin's affected area.

$$S_{en} = \frac{t_p}{t_p + t_n} \qquad (4)$$

The sensitivity of the developed AGEbRF technique is calculated and validated using popular methods such as the HFF, MTE, SCC, ELT, DSC, and RCSC approaches described in the Table 3.

Existing approaches have only achieved a sensitivity of nearly 95 percent. Furthermore, this AGEbRF method achieved 99.34 percent higher sensitivity than other methods, demonstrating the effectiveness of the developed model. Figure 9 depicts the sensitivity comparison graphically.

### Specificity measurement

Specificity is defined as the degree used to determine the number of true negatives that are correctly recognized. Additionally, specificity is used to calculate the efficiency of predicting skin cancer using segmented images, which is calculated using Eq. (5),

$$S_{pe} = \frac{t_n}{t_p + t_n} \qquad (5)$$

The specificity of the developed AGEbRF technique is calculated and validated using commonly used methods such as HFF, SCC, MTE, ELT, DSC, and RCSC approaches, which yielded values as shown in Table 4.

**Table 2  Validation of accuracy.**

| No. of samples | Accuracy (%) | | | | | |
| --- | --- | --- | --- | --- | --- | --- |
| | HFF | SCC | MTE | DSC | RCSC | Proposed |
| 10 | 99.85 | 86.9 | 92.79 | 86.12 | 94.8 | 99.96 |
| 20 | 98.12 | 85.67 | 90.05 | 84.2 | 93.3 | 99.54 |
| 30 | 96.7 | 84 | 89.43 | 82.22 | 91.78 | 99 |
| 40 | 94.32 | 83.3 | 87.7 | 80 | 90 | 98.66 |
| 50 | 92 | 81.29 | 85 | 78.29 | 89.12 | 98.12 |

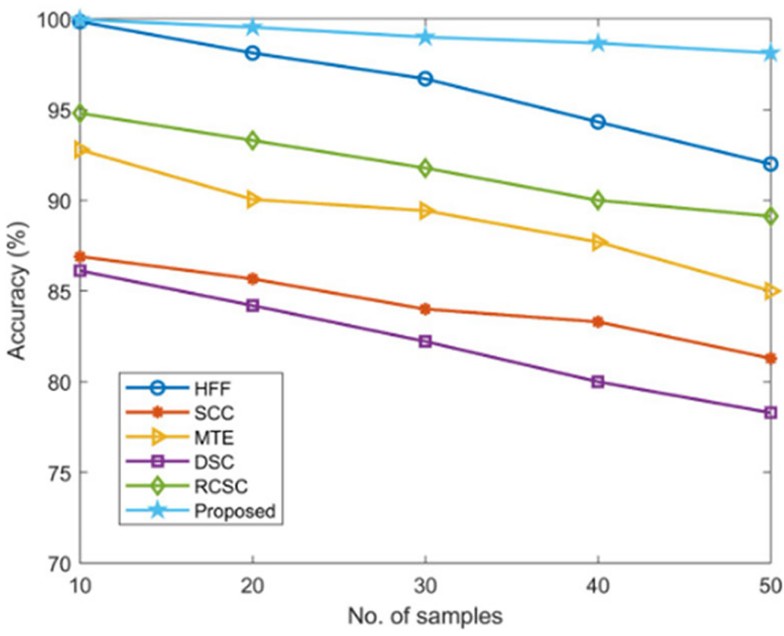

**Figure 8  Comparison of accuracy.**               

The existing approaches achieved a lower specificity of nearly 98 percent. As a result, the developed approach achieved 99.55 percent higher specificity than other methods, demonstrating the effectiveness of the developed model. In addition, Figure 10 depicts the comparison of specificity.

### Precision measurement

This procedure has been tested for determining the number of correct positive estimates alienated by the total positive estimates. Furthermore, precision is the proportion of precise cancer-affected region diagnosis computed using Eq. (6),

$$P_{re} = \frac{t_p}{t_p + f_p} \tag{6}$$

Using established techniques including HFF, SCC, MTE, ELT, and RCSC approaches, the suggested AGEbRF model's precision is estimated and validated, and the results are

**Table 3 Validation of sensitivity.**

| No. of samples | Sensitivity (%) | | | | | | |
|---|---|---|---|---|---|---|---|
| | HFF | MTE | SCC | ELT | RCSC | DSC | Proposed |
| 10 | 91.65 | 90.99 | 86.14 | 93.4 | 94.5 | 85.31 | 99.34 |
| 20 | 88.3 | 87.60 | 84.34 | 92 | 93.2 | 83.02 | 99.12 |
| 30 | 85.43 | 85 | 83.03 | 91.02 | 91.03 | 81 | 98.67 |
| 40 | 83 | 82.82 | 81.89 | 89.02 | 90 | 79.93 | 98 |
| 50 | 81.10 | 79.32 | 80 | 88 | 88.73 | 77.7 | 97.26 |

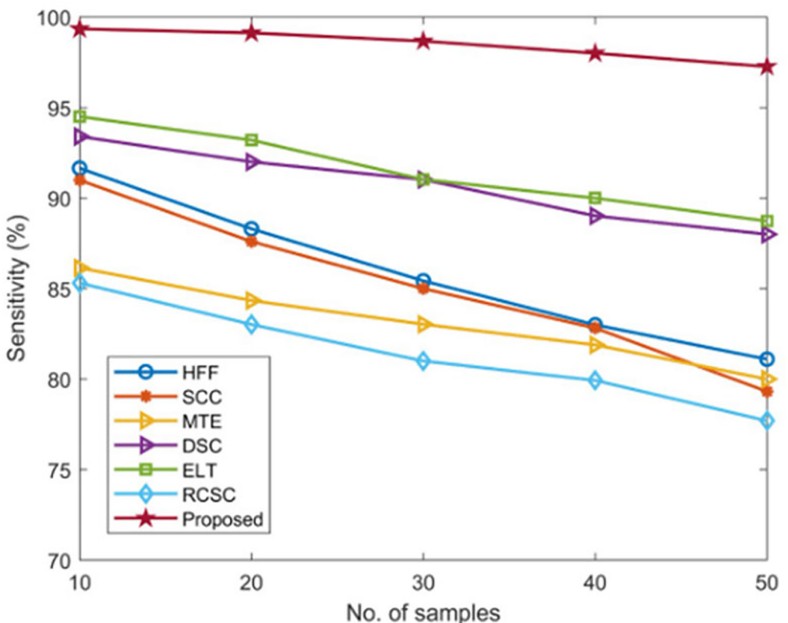

**Figure 9 Comparison of sensitivity.**

**Table 4 Validation of specificity.**

| No. of samples | Specificity (%) | | | | | | |
|---|---|---|---|---|---|---|---|
| | HFF | SCC | MTE | ELT | RCSC | DSC | Proposed |
| 10 | 95.70 | 87.66 | 89.19 | 97.4 | 98 | 84.5 | 99.55 |
| 20 | 93 | 85.6 | 87 | 96.5 | 97.45 | 82.53 | 99.12 |
| 30 | 91.15 | 84.89 | 84.83 | 94.44 | 96.66 | 80 | 98.77 |
| 40 | 89 | 82.3 | 82.15 | 93.12 | 94.87 | 78.31 | 98.12 |
| 50 | 87.92 | 80 | 80.05 | 92 | 93 | 77.43 | 98 |

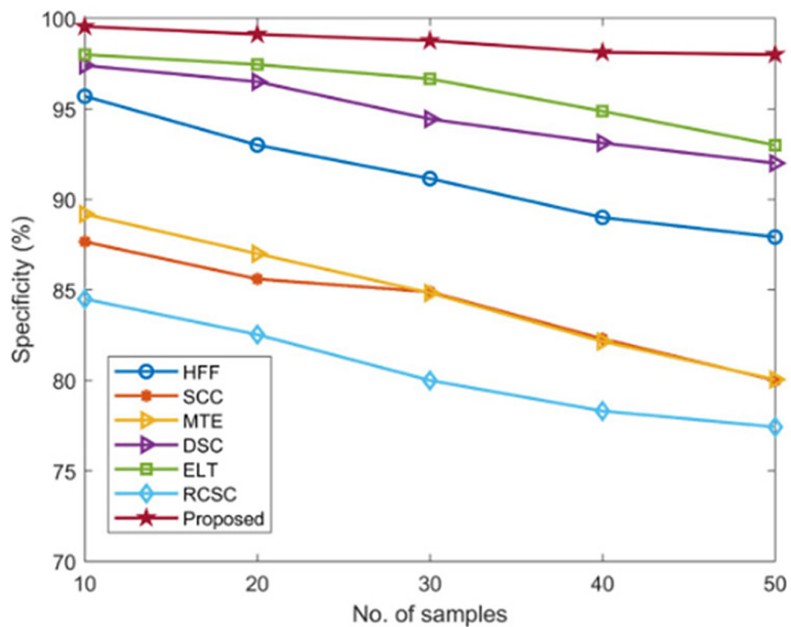

**Figure 10 Comparison of specificity.**     

**Table 5 Validation of precision.**

| No. of samples | Precision (%) | | | | | |
|---|---|---|---|---|---|---|
| | HFF | SCC | MTE | ELT | RCSC | Proposed |
| 10 | 95.60 | 87.47 | 88 | 85.9 | 95 | 99.29 |
| 20 | 93.24 | 86.21 | 86.12 | 84.3 | 94.3 | 99 |
| 30 | 92.13 | 84.87 | 83 | 82.03 | 93.35 | 98.87 |
| 40 | 91.2 | 83.3 | 81.23 | 80.45 | 91.12 | 98.34 |
| 50 | 90 | 82 | 79.42 | 78.8 | 89.4 | 98 |

listed in Table 5. Here, the current methods have only managed a lower precision of roughly 96 percent. Additionally, the created AGEbRF methodology obtained a 99.29% higher precision value than other methods and the comparison of precision is shown in Figure 11.

### F1-score measurement

The computation is based on measurements of sensitivity and precision used to determine how effectively the images are traced using an Eq. (7),

$$F1\text{--}score = 2\left(\frac{P_{re} * S_{en}}{P_{re} + S_{en}}\right) \tag{7}$$

where $P_{re}$ denotes the calculated precision value and $S_{en}$ represents the calculated sensitivity value.

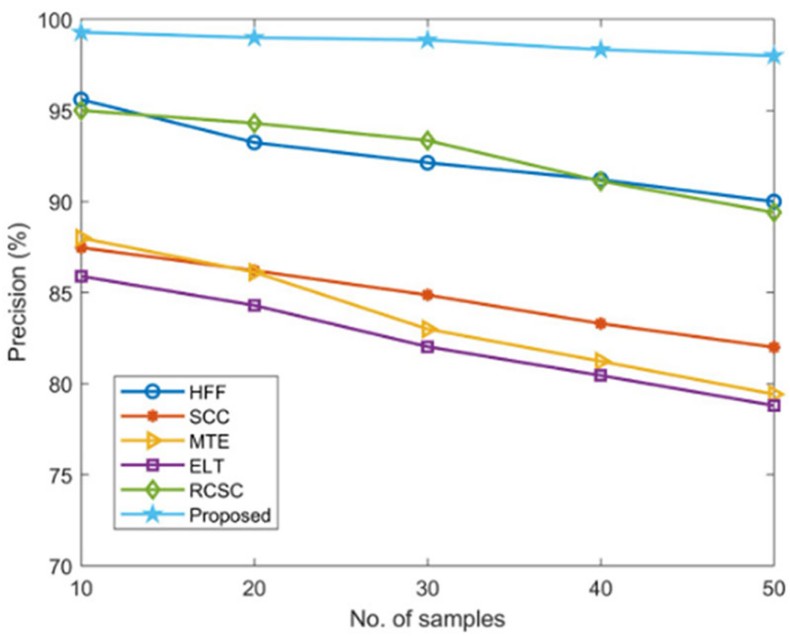

**Figure 11 Comparison of precision.**

**Table 6 Validation of F1-score.**

| No. of samples | F1-score (%) | | | | | |
|---|---|---|---|---|---|---|
| | **HFF** | **SCC** | **MTE** | **DSC** | **RCSC** | **Proposed** |
| 10 | 92 | 86.12 | 91.2 | 82 | 94.75 | 99.12 |
| 20 | 89.32 | 85.5 | 88.12 | 81.04 | 93 | 98.56 |
| 30 | 87 | 84.23 | 86.74 | 78.76 | 92.13 | 98 |
| 40 | 85.52 | 83 | 82.23 | 76 | 91.78 | 97.5 |
| 50 | 83 | 82.30 | 79.42 | 74.32 | 89.02 | 97 |

The suggested AGEbRF model's F1-score is determined and validated using widely used techniques such as hybrid feature fusion and ML (HFF) (*Rahman et al., 2022*), SCC (*Mijwil, 2021*), MTE (*Wei et al., 2021*), and DSC (*Murugan et al., 2021*), and RCSC (*Saba et al., 2019*) approaches. The results are listed in the Table 6.

Here, the suggested model has outperformed previous methods with a high F1-score value of 99.12% compared to the existing approaches, as shown in Figure 12.

## RESULTS AND DISCUSSION

Python is used to process the created AGEbRF replica, and the predicted model's success rate is evaluated using the mechanisms that are now in place in terms of accuracy, sensitivity, F-measure, precision, and specificity. Dermoscopic images are used in this method to find skin cancer. The suggested AGEbRF technique identifies skin cancer; in this case, using dermoscopic images divides the affected skin cancer portions. As a result, the generated model performed well in segmentation and prediction.

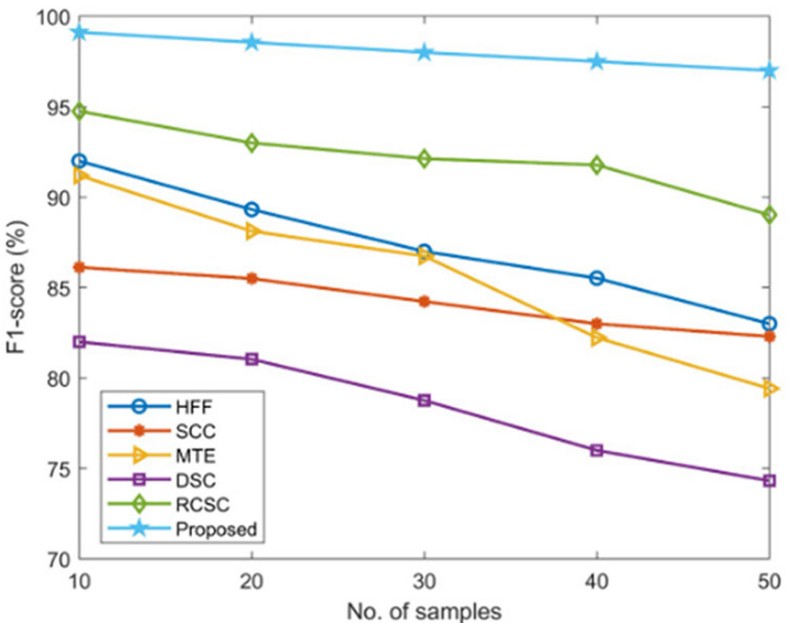

**Figure 12 Comparison of F1-score.**   

**Table 7 Overall performance metrics.**

| Methods | Performance assessment with key metrics | | | | |
| --- | --- | --- | --- | --- | --- |
| | Accuracy | precision | sensitivity | specificity | F-measure |
| HFF | 99.85 | 95.60 | 91.65 | 95.70 | 92 |
| SCC | 86.90 | 87.47 | 86.14 | 87.66 | 86.12 |
| MTE | 92.79 | 88 | 90.99 | 89.19 | 91.2 |
| RCSC | 94.8 | 95 | 94.5 | 98 | 94.75 |
| Proposed | 99.96 | 99.29 | 99.34 | 99.55 | 99.12 |

By achieving the best results in accuracy, sensitivity, specificity, F-measure, and precision, the suggested model of AGEbRF has demonstrated good performance. As a result, the devised approach fixed the training problems at the beginning. Extraction of characteristics based on color, shape, size, and texture features is the next step. Additionally, segmentation is carried out in the classification layer and predicts the melanoma's malignancy or benignity in the output layer. As a result, the newly developed AGEbRF technology improves the performance of predicting affected skin portions.

The outstanding metrics comparisons are listed in Table 7 involves the comparison of HFF (*Rahman et al., 2022*), SCC (*Mijwil, 2021*), MTE (*Wei et al., 2021*), DSC (*Murugan et al., 2021*), and RCSC (*Saba et al., 2019*) approaches, and the suggested AGEbRF had the best results across every parameter validation. Furthermore, the created framework improved in terms of predicting accuracy, sensitivity, and specificity, each of which reached 99.55 percent. Thus, the proposed AGEbRF robustness is confirmed, and it can divide the skin's affected areas and identify its stages, such as benign or malignant. To

precisely segment the part of the body affected by skin cancer, the proposed AGEbRF technique is used. Here, utilizing dermoscopic images, the suggested method has demonstrated great performance for segmenting and predicting processes in skin cancer.

## CONCLUSIONS

This study develops a novel AGEbRF approach for detecting and segmenting skin cancer-affected areas. The dermoscopic images are used as input images for training the system. Pre-processing, feature extraction, classification, segmentation, and prediction are all part of the proposed AGEbRF model. Furthermore, the developed model is applied to a dermoscopic image dataset, and the affected areas of the skin cancer are segmented. Furthermore, the segmented images are used to detect the presence of skin cancer. The proposed model achieved better accuracy, sensitivity, specificity, and precision results. When compared to existing models, it achieved 99.96 percent accuracy in predicting skin cancer-affected parts. Still the future works should concentrate on unbiased data and to check if these un-biasing procedures would create an impact in the results obtained and to work on the updated datasets to compare the performance in several versions of datasets and on different tasks.

### Funding
The authors received no funding for this work.

### Competing Interests
Rajanikanth Aluvalu is an Academic Editor for PeerJ.

### Author Contributions
- Prasanalakshmi Balaji conceived and designed the experiments, analyzed the data, performed the computation work, prepared figures and/or tables, and approved the final draft.
- Bui Thanh Hung analyzed the data, authored or reviewed drafts of the article, and approved the final draft.
- Prasun Chakrabarti performed the experiments, performed the computation work, prepared figures and/or tables, and approved the final draft.
- Tulika Chakrabarti performed the experiments, prepared figures and/or tables, authored or reviewed drafts of the article, and approved the final draft.
- Ahmed A. Elngar analyzed the data, authored or reviewed drafts of the article, and approved the final draft.
- Rajanikanth Aluvalu conceived and designed the experiments, performed the experiments, analyzed the data, performed the computation work, prepared figures and/or tables, authored or reviewed drafts of the article, and approved the final draft.

### Data Availability
The code is available in the Supplemental File and the raw data is available at figshare: Codella, Noel; Rotemberg, Veronica; Tschandl, Philipp; Emre Celebi, M.; Dusza, Stephen; Gutman, David; et al. (2023): ISIC2018_Task1-2_Training_Input.zip. figshare. Dataset. https://doi.org/10.6084/m9.figshare.22698379.v1.

The data used for this work included the HAM10000 dataset and the ISIC challenge Datasets 2018: https://challenge.isic-archive.com/data/#2018.

## Supplemental Information

Supplemental information for this article can be found online at http://dx.doi.org/10.7717/peerj-cs.1387#supplemental-information.

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
