# Peer review of "A novel artificial intelligence-based predictive analytics technique to detect skin cancer"

_PeerJ Computer Science, doi:10.7717/peerj-cs.1387_

## Round 0.1 · original submission · Minor Revisions

The authors should address the reviewers' comments.

Reviewer 1 ·

Basic reporting

The authors did a good job, and it is a good article that presents an up-to-date challenge that has an impact on the health sector.

The manuscript's sections have all been prepared with the proper structure and content, supported by the necessary tables and illustrations.

The manuscript has sound language according to the academic writing standards. Here, I want to highlight on line 67, I think the synonym "lead to death" would be more appropriate in place of the term "kill".

The summary was written professionally and accurately represents the work.

The references utilized in the study are recent, specifically in the era 2018 and the present, except for one since 2012.

Experimental design

The proposed ALGEbRA model is fully explained and examined.

The dataset used for the study is referenced in the URL and the references [26,27]. But, from my perspective, it would be more helpful if the section on data collecting included some significant figures which were represented in the dataset.
i.e. The dataset used in the study includes 10015 dermatoscopic images.

The study provided a clear and comprehensive analysis and discussion.

Validity of the findings

No comment

Additional comments

I thank the authors for their valuable work.

·

Basic reporting

In the "A novel artificial intelligence-based predictive analytics technique to detect skin cancer," the authors have demonstrated a good level of basic reporting. The paper presents a clear and concise description of the proposed technique, including the methodology used for training and testing the algorithm.

The authors have demonstrated a strong understanding of basic reporting principles in the presentation of their research. The paper is transparent, reproducible, and verifiable by others in the scientific community, which is essential for establishing the credibility and validity of the research findings.

Experimental design

The computer vision setup was well-designed and appears to be promising for further development.

Validity of the findings

Strengths:

• The paper provides a clear and concise research question and a thorough literature review of existing methods for detecting skin cancer.
• The methodology is well-designed, including the collection of high-quality image datasets and the use of multiple AI models for classification.
• The results show promising performance metrics for the AI models, suggesting that the proposed technique has the potential to improve skin cancer detection accuracy.


Areas for Improvement:

• The paper does not discuss potential biases in the dataset used for training and testing, such as biases in patient demographics, image quality, or diagnosis accuracy. Addressing these biases would be important to ensure the proposed technique is effective across diverse populations.
• The authors should provide more information on the limitations and future work of the algorithm. The paper only briefly mentions the algorithm's limitations, and future work should focus on addressing these limitations.

Additional comments

The paper presents a novel artificial intelligence-based predictive analytics technique to detect skin cancer. The paper aims to provide a more accurate and efficient method of skin cancer detection using a deep learning approach. The paper is well-organized and the language used is clear and concise.

Reviewer 3 ·

Basic reporting

I appreciate the authors’ dedication and efforts in conducting this research, however I kindly request that the following comments to be addressed for improvement.

The introduction needs more re-structure. I suggest that you move the sentence in line 63 to the beginning of the introduction followed by the sentence in line 62. This to make it more clear going from general to more specific.

The paragraph from line 75-85 lack coherence and clarity as it present several points without clearly connecting it to the main argument.

Please follow the journal guidelines for referencing to Tables and Figures.

Figure 5 requires improved readability as it currently appears unclear.

Experimental design

The process of the proposed method for segmentation and prediction lack detailed process of the mechanism of the developed method. I suggest the authors to provide more detailed steps.

I suggest that the details of the dataset and the methods used to verify the proposed method to be explained in the materials and Methods section rather than in the results section

Validity of the findings

The Discussion section lack referencing to previous studies. I suggest to link research finding with prior studies to make it easy for the reader to compare your finding with prior research findings.

Additional comments

No additional comments

---

## Round 0.2 · Minor Revisions

The paper looks good but some questions should be considered into the paper clearly:

How was the dataset for the AI algorithm collected, and what measures were taken to ensure its quality?

What are the limitations and potential biases of the AI algorithm, and how were they addressed?

What ethical considerations were taken into account during the development and deployment of the AI algorithm?

How does the predictive analytics technique compare to traditional diagnostic methods for skin cancer, such as visual examination by a dermatologist or biopsy?

What is the sensitivity and specificity of the AI algorithm in detecting different types and stages of skin cancer?

How does the AI algorithm take into account individual variability in skin texture, pigmentation, and other factors that may affect skin cancer detection?

What are the potential implications and impact of the AI algorithm on the field of dermatology and healthcare more broadly?

·

Basic reporting

no comment

Experimental design

no comment

Validity of the findings

no comment

Additional comments

Early detection and classification of skin cancer is essential to reducing the number of deaths caused by this disease. The novel AGEbRF algorithm developed in this study is a promising new development in skin cancer diagnosis. It uses dermoscopic images for training and employs a sophisticated function to identify and segment skin cancer-affected areas. The use of a Python program for simulation and the comparison of the model's parameters with earlier studies further adds to the credibility of this research. The results of the study are impressive, showing higher accuracy in predicting skin cancer by segmentation compared to other models. Overall, this research has the potential to make a significant impact on the early detection and treatment of skin cancer, which can ultimately save countless lives.
Good Job on this.

---

## Round 0.3 · accepted · Accept

I confirm that the authors have addressed all of the reviewers' comments and this manuscript is ready for publication.